# Study on the Suitability of Tea Cultivars for Processing Oolong Tea from the Perspective of Aroma Based on Olfactory Sensory, Electronic Nose, and GC-MS Data Correlation Analysis

**DOI:** 10.3390/foods11182880

**Published:** 2022-09-16

**Authors:** Chang He, Yuchuan Li, Jingtao Zhou, Xinlei Yu, De Zhang, Yuqiong Chen, Dejiang Ni, Zhi Yu

**Affiliations:** 1Key Laboratory of Horticulture Plant Biology (Ministry of Education), College of Horticulture & Forestry Sciences, Huazhong Agricultural University, Wuhan 430070, China; 2Key Laboratory of Urban Agriculture in Central China, Ministry of Agriculture, Wuhan 430070, China; 3Hubei Hongshan Laboratory, Wuhan 430070, China

**Keywords:** oolong tea, cultivars, olfactory sensory, electronic nose, GC-MS, suitability

## Abstract

The oolong tea aroma is shown to consist of cultivar aroma and technical aroma in this study based on the aroma differences between oolong tea products of cultivars of different suitability, as determined by correlation analysis of olfactory, sensory, electronic nose, and GC-MS data. Human senses were significantly affected by the aroma components, which included eight terpene metabolites (β-Ocimene, (Z)-Furan linalool oxide, linalool, (3E)-4,8-Dimethyl-1,3,7-nonatriene, (E)-Pyranoid linalool oxide, γ-Elemene, Humulene, (Z,E)-α-Farnesene), three carotenoid metabolites (β-Ionone, (Z)-Geranylacetone and 6-methyl-5-Hepten -2-one), three lipid metabolites ((Z)-3-Hexenyl (Z)-3-hexenoate, Butanoic acid hexyl ester, and (Z)-Jasmone), four amino acid metabolites (Methyl salicylate, Geranyl isovalerate, indole, and Phenylethyl alcohol), and six thermal reaction products (2-Pentylfuran, Octanal, Decanal, (E,E)-2,4-Nonadienal, (Z)-2-Decenal, and (E)-2-Undecenal). Meanwhile, several aroma compounds (such as (E)-Nerolidol and α-Farnesene), mainly comprising the “technical aroma” formed in the processing mode, were noted to be less closely related to cultivar suitability. This study sheds light on the aroma characteristics of different tea cultivars for oolong tea processing.

## 1. Introduction

Oolong tea is mainly produced in China, especially in South China, and it is loved by consumers for its unique antioxidant, anti-inflammatory, anti-cancer, floral, and fruity aroma qualities [1]. Aroma is the main quality factor influencing the market price of oolong tea, and oolong tea with a more obvious floral and fruity aroma is of superior quality and higher price [2]. The main factors affecting the aroma quality of oolong tea include tea cultivars, agronomic techniques, and processing techniques, with tea cultivar as the most important influencing factor [3].

In China, not all tea cultivars are suitable for oolong tea production. Previous studies on the suitability of oolong tea cultivars have mainly focused on the differences in the morphology and chemical composition of tea leaves. In one study, eight morphological and anatomical features were reported to be among the characteristics of suitable oolong tea cultivars, such as broad leaves, thick stems, and short internodes [4]. In another study, suitable oolong tea cultivars were shown to have higher levels of free amino acids and lower soluble sugar levels in young fresh leaves, while having higher caffeine content and lower tea polyphenol content in mature fresh leaves [5]. However, there are rare reports on the aroma characteristics of tea cultivars suitable for oolong tea production. A recent study compared changes in the volatile content of the suitable cultivar “foshou” and the unsuitable cultivar “Zhenong 139” during tea processing, and “foshou” was found to have a higher ratio of terpenoid volatiles (TVs) and green leaf volatiles (GLVs) than “Zhenong 139”, so the ratio of TVs to GLVs was speculated to be related to the cultivar suitability of oolong tea and to have a potential effect on the chemical basis of the oolong tea aroma characteristics [3]. However, in their study, only one suitable oolong tea cultivar was used, suggesting insufficient evidence. Thus far, whether tea cultivars are suitable for making oolong tea has been judged mainly by sensory experts. The influence of tea cultivars on oolong tea aroma formation has rarely been studied, and the specific aroma components affecting the suitability are not often mentioned. This suggests the need to explore the possibility of judging the suitability of oolong tea cultivars through aroma analysis of different cultivars, with a focus on the contribution of aroma components to the suitability of oolong tea cultivars.

The purpose of this study was to explore the suitability of oolong tea cultivars from the aroma perspective through correlation analysis of olfactory sensory, electronic nose, and GC-MS data based on the manufacturing process of “qingxiang” oolong tea, and using three suitable and three unsuitable oolong tea cultivars as the research objects. The sensory attributes of tea were evaluated by experienced experts according to widely used standards (such as Chinese national standards) [6]. Additionally, consumer evaluation is often used to evaluate food sensory attributes and predict the potential economic value of products [7,8]. As a consumer product, consumer evaluation of tea is less often mentioned in tea research. In this study, a combination of expert evaluation and consumer evaluation was conducted to analyze the differences in aroma sensory attributes to determine the suitability of the selected tea cultivars. Electronic nose can simulate human senses and is often used to identify the aroma characteristics of food [9]. For example, electronic nose was used to identify the classification of Indian black tea [10]; electronic nose combined with gas chromatography and mass spectrometry (GC-MS) was used to monitor the aroma formation process of chestnut-flavored green tea [11]. A study reported that a change in pyridine content in coffee bean roasting was detected by electronic nose and SPME-GC-MS [12]. In addition, the degradation changes in rapeseed oil under prolonged heating were determined by electronic nose combined with SPME-GC-MS technique [13] In this study, electronic nose was used to distinguish the differences in the aroma of samples from different tea cultivars, thus obtaining more specific evidence of suitability classification. Moreover, headspace-solid phase microextraction (HS-SPME) combined with GC-MS, a technology widely used in the research of aroma compounds [14], was also used in this study. As electronic nose and HS-SPME-GC-MS results are related to sensory properties, correlation analysis between them was applied to further explore the differences in aroma compounds and their effects on the suitability of oolong tea cultivars.

## 2. Materials and Methods

### 2.1. Plant Materials and Manufacturing Process of Oolong Tea

In this study, six tea cultivars were selected [15], including three suitable cultivars, namely Camellia sinensis (L.) O. Kuntze cv. Tie guanyin (TGY, No. GS13007-1985, recognized as suitable for making oolong tea and green tea by the Chinese Crop Varieties Examining Committee in 1985); Camellia sinensis (L.) O. Kuntze cv. Meizhan (MZ, No. GS13004-1985, recognized as suitable for making oolong tea, green tea, and black tea by the Chinese Crop Varieties Examining Committee in 1985); Camellia sinensis (L.) O. Kuntze cv. Huangdan (HD, No. GS13008-1985, recognized as suitable for making oolong tea, green tea, black tea by the Chinese Crop Varieties Examining Committee in 1985); and three unsuitable cultivars, namely Camellia sinensis (L.) O. Kuntze cv. Fuding Dabaicha (FD, No. GS13001-1985, suitable for making green tea, black tea, and white tea as determined by the Chinese Crop Varieties Examining Committee); Camellia sinensis (L.) O. Kuntze cv. Yingshuang (YS, No. GS13041-1987, recognized as suitable for making black tea and green tea by the Chinese Crop Varieties Examining Committee in 1987); and Camellia sinensis (L.) O. Kuntze cv. Jiaming 1 (WNZ, also known as Wu Niu Zao, No. Zhepin Lizi No. 079, recognized as suitable for making green tea by the Zhejiang Province Crop Varieties Examining Committee in 1988). The resting bud and two or three leaves of the six tea cultivars were collected in May, and the manufacture of oolong tea followed the general method for “qingxiang” Tieguanyin [16]. Briefly, the freshly plucked tea leaves were withered indoors for 2 h, followed by exposure to sunlight (leaf temperature 32 °C, 24000 Lux) for 30 min, and weathering indoors again for 2 h at 25 °C and 70% relative humidity. Next, the tea leaves were turned over (20 rpm) for 2 min, followed by withering for 90 min, turning over (20 rpm) again for 4 min, withering for 180 min, turning over (20 rpm) again for 6 min, and further withering for 360 min. After withering, the tea leaves were fixed at 220 °C in a fixing machine (NO. 6CST-50, Quanzhou, China), followed by rolling for 10 min in a roller machine (NO. 6CR-25, Quanzhou, China), and drying at 110 °C for 8 min in a dryer machine (NO. 6CTH-60, Quanzhou, China). Finally, the tea leaves of the six cultivars were completely dried (moisture content less than 6%) at 80 °C, and the dried tea samples were stored at −20 °C for further analysis.

### 2.2. Aroma Quality Assessment by Experts

The aroma quality of the six oolong tea samples was evaluated three times by three experienced tea experts based on the Chinese national standard procedure for tea leaf evaluation [6]. Briefly, each sample (5 g) was brewed with boiling water in a gaiwan (110 mL, including a bowl and a lid) for 1 min, followed by removal of the lid and evaluation of the aroma of each sample for the first time; 2 min later, the tea soup was poured out, followed by brewing of the samples left in the gaiwan again with boiling water for 2 min, removal of the lid, and evaluation of the aroma of each sample for the second time. One min later, the tea soup was poured out, and the samples left in the gaiwan were brewed with boiling water for 3 min, followed by removal of the lid and evaluation of the aroma of each sample for the third time. The aroma results were based on the three evaluation results, with a focus on the second aroma evaluation.

### 2.3. Acceptance Assessment by Consumers

The aroma acceptance and purchase intention were evaluated for the six oolong tea samples by 63 oolong tea consumers (aged 18 to 42), and 53 of them had consumed oolong tea products one week before the experiment. Each consumer was asked to evaluate the six oolong tea products in a separate compartment of the flavor review laboratory. Consumers were given water to cleanse their olfactory system between samples. Consumers were initially instructed about the evaluation task and the use of the scoring scale in this research. The data for the consumers were collected and stored in accordance with Huazhong Agricultural University Human Ethics application ID Number: HZAUHU-2020-0017.

Briefly, each sample (3 g) was brewed with 150 mL boiling water for 5 min in a cylindrical cup, followed by the pouring of the tea soup into a matched bowl, and the random numbering of the cylindrical cup and the matched bowl for blind review. A nine-point scale was used for evaluation (with 1 point for the lowest evaluation and 9 points as the highest evaluation) in terms of aroma, soup color, taste, infused leaf, and overall acceptance of oolong tea products. Finally, the product purchase intention of the consumers was judged using a five-point scale (1 point for no purchase intention; 2 points for little purchase intent; 3 points for an approximate purchase intention; 4 points for a lot of purchase intent; 5 points for a strong purchase intent).

### 2.4. Electronic Nose Detection

The aroma characteristics of the 6 samples were measured using the FOX 4000 nose (Alpha-MOS, Toulouse, France), equipped with 18 metal oxide sensors (LY2/LG, LY2/G, LY2/AA, LY2/GH, LY2/gCTL, LY2/gCT, T30/1, P10/1, P10/2, P40/1, T70/2, PA/2, P30/1, P40/2, P30/2, T40/2, T40/1, TA/2) and a headspace auto sampler HS100. Briefly, each sample (2 g) was crushed by liquid nitrogen and brewed with 5 mL boiling water in a 20 mL headspace vial, followed by maintenance of the headspace vial at 60 °C for 20 min under agitation (500 rpm) and placing of the sample in the electronic nose injector. Headspace (2500 mL) carried by dry air (150 mL/min) was injected into the E-nose at an injected temperature of 60 °C and an injected volume of 2500 µL. Sensor resistance was measured at one acquisition per second for 120 s. Alpha SOFTV9.1 software was used to analyze the obtained data, and the stable response value of each sensor was extracted as the electronic nose data of each sample.

### 2.5. Extraction and Analysis of Volatile Compounds in Oolong Tea

Volatile profiles were directly collected from each sample with a modified headspace-solid phase microextraction (HS-SPME) method and detected by TRACE-DSQ-Ⅱ GC–MS (Thermo Fisher, Waltham, MA, USA) as previously reported [17]. Briefly, each sample (2 g) crushed by liquid nitrogen was brewed with 5 mL boiling water in a 20 mL headspace vial, followed by the addition of 20 μL (4 μL/100 mL) of ethyl decanoate as an internal standard (IS) and immediate sealing of the vial. After fixing the polydimethylsiloxane/divinylbenzene (PDMS/DVB) fiber (Supelco, Bellefonte, PA, USA) on its upper end, the headspace vial was kept at 60 °C for 60 min to absorb the volatile gases. The analysis was performed under the following conditions: chromatographic column, DB-5MS (30 mm × 0.25 mm × 0.22 μm); inlet temperature, 230 °C; carrier gas, high-purity helium (purity ≥ 99.99%); column flow rate, 1.0 mL/min; heating program, an initial temperature of 40 °C and hold for 2 min, up to 85 °C at 5 °C/min and hold for 2 min, up to 110 °C at 2 °C/min, up to 130 °C at 7 °C/min, up to 230 °C at 5 °C/min and hold for 8 min; oven temperature, 40 °C; injection mode, splitless. Ion source (EI) electron energy, 70 eV; ion source temperature, 230 °C; mass scanning range, m/z 35–400. Volatiles were characterized based on the mass spectral database and retention index (RI, determined by N-alkanes C3–C25) of the National Institute of Standards and Technology (NIST, Gaithersburg, MD, USA). The relative concentration of the detected volatiles was calculated by Equation (1):Relative concentration (μg/kg) = (Peak area of target/Peak area of IS) × 0.1726 μg/Amount of sample (g) × 1000(1)

### 2.6. Data Processing

Multiple regression analysis of the acceptance evaluation results of oolong tea samples and logistic analysis of the purchase intention evaluation results were performed using SAS 3.0 software (for details, refer to Appendix A). For data analysis, R 4.1.3 and R studio were used for Student t-test, cluster analysis, Kruskal–Wallis test, Spearman’s correlation analysis, PCA analysis, PCoA analysis, Adonis test and plotting Figure 1, Figure 2 and Figure 3. Gephi 0.9.2 was used to draw Figure 4. The significant differences between different groups were analyzed using the specific methods shown in the related figures and tables, unless stated otherwise.

## 3. Results and Discussion

### 3.1. Olfactory Sensory Evaluation of Different Oolong Tea Samples

The expert evaluation results and the consumer acceptance evaluation results are shown in Figure 1 and Appendix A. In Figure 1A–D, the scores of both experts and consumers are shown to be significantly higher (*p <* 0.01) for the suitable tea cultivars (MZ, HD and TGY) than the unsuitable cultivars (FD, WNZ, and YS). Based on expert evaluation, the products of the suitable cultivars had a stronger and richer floral and fruity aroma than those of the unsuitable cultivars (Appendix A), and the floral and fruity aroma of Oolong tea was not influenced by different producing regions [18]. Based on the evaluation of consumers, the acceptance of tea products may be influenced by multiple factors apart from aroma (Appendix A). Multiple regression analysis (Appendix A) and logistic regression analysis (Appendix A) indicated that among several common factors influencing consumer acceptance, aroma is always an important factor, even with the interaction of different factors (Appendix A). In the acceptance evaluation, the consumers’ reviews of the aromas of different oolong tea products were comprehensive and objective, with high scores from both the expert evaluation and consumer evaluation for the products of the suitable cultivars, reflecting the economic value and significance of the application of suitable oolong tea cultivars.

### 3.2. Electronic Nose Detection of Different Oolong Tea Samples

In Figure 2 and Appendix A, the electronic nose results showed significant differences (*p <* 0.05) between the samples of suitable and unsuitable cultivars. In the heat map (Figure 2A), all samples were divided into two major categories by cluster analysis, agreeing with the traditional suitability classification of oolong tea cultivars. LY-type sensors (LY2/AA, LY2/gCTL, LY2/gCT, LY2/G, and LY2/GH) had positive response values to the suitable tea cultivars (MZ, HD, and TGY) and negative response values to the unsuitable tea cultivars (FD, WNZ, and YS). Unlike the 5 LY-type sensors, the P-type sensors (P30/2, P10/2, P30/1, P40/1, P40/2, P10/1, and PA/2) and T-type sensors (T40/2, T30/1, and T70/2) showed intergroup differences in their response values to the six different samples. Based on the average response values of sensors, cluster analysis divided the sensors into three categories, with 15 sensors distinguishing the suitability grouping of oolong tea cultivars at *p <* 0.01 (Figure 2A, *p* value, Kruskal–Wallis test, with *p <* 0.05 for TA2), excluding T40/1 and LY2/LG due to *p* > 0.05.

Moreover, the normalized data (z-score normalization) of the electronic nose sensor response values were further examined by PCA analysis, and the score map also showed the data difference in the two groups (Figure 2B). In the score chart, the sum contribution rates of principal components 1 (*x*-axis) and 2 (*y*-axis) to the overall model exceeded 90%, and the scores of the suitable tea cultivars were significantly different (*p <* 0.05) from those of the unsuitable cultivars in principal component 1 (92.15%) on the distribution (*x*-axis). Meanwhile, in Appendix A, the Adonis analysis results show a significant difference (R^2^ = 0.5707 and *p <* 0.01) between the samples of different suitability, further verifying the reliability of sample suitability grouping.

Electronic nose, a sensor system simulating the human nose, can quickly identify tea aroma differences [19]. In this study, the electronic nose was able to significantly distinguish the products of suitable and unsuitable oolong tea cultivars, and all the data were statistically significant (*p <* 0.05). Not all sensors played a significant role in this process, so we excluded T40/1 and LY2/LG in the subsequent analysis. Due to the differences in the contents of aroma substances, oolong tea products from cultivars of different suitability varied in the response values of the electronic nose sensors, and the tea products in the same suitability group showed certain similarities in their aroma characteristics. Furthermore, the electronic nose data were consistent with the sensory review data, which not only reflected the accuracy of electronic nose detection, but also the close relationship between the electronic nose data and human senses (sensory review data). Both datasets were derived from the aroma components of different samples.

### 3.3. Volatile Detection of Different Oolong Tea Samples

HS-SPME-GC-MS results showed that the six oolong tea samples varied in aroma compounds. In Figure 3A,B, the total number of aroma substances are shown to be significantly higher (*p <* 0.05) in the products of the suitable tea cultivars (MZ, HD, and TGY) than in the products of the unsuitable tea cultivars (FD, WNZ, YS). Among them, HD had a significantly higher (*p <* 0.01) aroma level than the other cultivars, while YS showed the lowest aroma, indicating that the suitability of the oolong tea cultivars affected the total aroma. Statistical analysis of aroma compounds in different metabolic pathways [20] showed that volatile terpenes (VTs) dominated the aroma (more than 60%) in all samples, followed by fatty-acid-derived volatiles (FADVs) and amino-acid-derived volatiles (AADVs) (both over 10%), carotenoid-derived volatiles (CDVs) (about 5%), and some other unclassified substances (others).

Furthermore, we compared the content data of each aroma compound in different samples (Figure 3D and Appendix A). As the aroma data of each sample varied in volatile substances, based on suitability grouping, the non-parametric test (Kruskal–Wallis test) showed the clear difference between suitable and unsuitable tea cultivars (MZ, HD, and TGY versus FD, WNZ, and YS) in aroma compounds. In Figure 3D, the 64 substances are presented on the right side according to the classification of metabolic pathways, with clustering results based on the relative content of substances shown in the upper part of the heat map, which also confirms the PCoA classification results.

For volatile terpenes (VTs), the two groups showed an extremely significant difference (*p <* 0.01) in compounds, including β-Ocimene, (Z)-Furan linalool oxide, (E)-Furan linalool oxide, linalool, (3E)-4,8-Dimethyl-1,3,7-nonatriene, Cosmene, (E)-Pyranoid linalool oxide, Linalool oxide pyranoside, α-Terpineol, linalyl formate, α-Cubebene, γ-Elemene, Humulene, (E)-β-Famesene, (Z,E)-α-Farnesene, α-Muurolene, α-Farnesene, T-Muurolol, and α-Cadinol, and a significant difference (*p <* 0.05) in compounds such as α-Terpineol, Geraniol, β-Caryophyllen, δ-Cadinene, and (E)-Nerolidol, while showing no significant difference (p > 0.05) in compounds such as 1,5,7-Octatrien-3-ol, 3,7-dimethyl, γ-Cadinene, Nerolidol, and (E)-2,6-Dimethyl-3,7-octadiene-2,6-diol. Linalool and its furan oxides ((Z)-Furan linalool oxide, (E)-Furan linalool oxide) are important components of sweet and floral aromas, while (E)-Pyranoid linalool oxide and Linalool oxide pyranoside have earthy aromas [21]. Geraniol is one of the main components of the floral odor in tea [22]. (E)-Nerolidol has floral green, citrus woody, and waxy odors, which are among the characteristic aromas of oolong tea [23]. The contents of these terpenes with floral aromas are higher in the samples of suitable oolong tea cultivars, suggesting that these compounds may affect the suitability of oolong tea cultivars and play a major role in the formation of the floral aroma.

With regard to amino acid derived volatiles (AADVs), the two groups showed extremely significant differences (*p <* 0.01) in compounds including Benzaldehyde, Phenylethyl alcohol, Methyl salicylate, Indole, (Z)-3-Hexenyl benzoate, and Geranyl isovalerate, and a significant difference (*p <* 0.05) in Phenylacetaldehyde, while showing no significant difference (*p* > 0.05) in compounds such as Benzyl nitrile, β-Phenylethyl butyrate, Benzoic acid hexyl ester, and E-2-Hexenyl benzoate. Phenylethyl alcohol and Phenylacetaldehyde have floral and sweet odors, Benzaldehyde has an almond odor, indole is one of the characteristic aroma substances of oolong tea [23] with a sour fruity odor, and (Z)-3-Hexenyl benzoate has a pear-like odor. These aroma compounds may contribute to the strong floral and fruity aromas in the products of suitable oolong tea cultivars.

With regard to fatty acid derived volatiles (FADVs), the two groups exhibited extremely significant differences (*p <* 0.01) in compounds such as Butanoic acid hexyl ester, (Z)-3-Hexenyl hexanoate, (Z)-3-Hexenyl (Z)-3-hexenoate, and (Z)-Jasmone, while showing no difference (*p* > 0.05) in 1-Heptanol, 1-Octen-3-ol, trans,(E)-2,4-Heptadienal, (Z)-3-Hexenyl-α-methylbutyrate, (Z)-3-Hexenyl isovalerate, Hexanoic acid hexyl ester, and (E)-2-Hexenyl. All the four significantly different metabolites have a fruity odor, with (Z)-3-Hexenyl hexanoate and (Z)-Jasmone as the main compounds in lipid metabolism due to their high content. Among the aroma compounds with no differences between the two groups, hexanoic acid hexyl ester, (E)-2-Hexenyl hexanoate and (Z)-3-Hexenyl-α-methyl butyrate all have a green odor. These results indicate that these different aroma compounds in FADVs may be among the factors responsible for the fruity aroma differences involved in the suitability of oolong tea cultivars.

Regarding carotenoid derived volatiles (CDVs), the two groups had extremely significant differences (*p <* 0.01) in the compounds of β-Ionone (woody, violet), (Z)-Geranyl acetone (Floral, hay-like), 6-methyl-5-hepten-2-one, and (r,s)-5-ethyl- 6-methyl-3e-hepten-2-one, and a significant difference (*p <* 0.05) in α-ionone (woody, hay-like). Among these compounds, β-ionone and (z)-geranyl acetone had a higher content with a floral and woody aroma. β-ionone was considered to contribute significantly to the black tea aroma [21] as a carotenoid metabolite in black tea. These aroma compounds can be assumed to have greatly affected the differences between woody and floral odors in the samples.

In this study, the two groups were also found to differ extremely significantly (*p <* 0.01) in the aroma compounds of 2-pentylfuran, octanal, decanal, (e, e)-2,4-nonadienal, (z)-2-decenal, methyl nerate, e-2-undecenal, and (e)-2-dodecenal, but with no significant differences (*p* > 0.05) in dodecane and isoeugenol. 2-pentylfuran has a bean-like, fruity, and green odor, which may be produced in the process of fixing or drying as a thermal reaction product [24]. Alkenal compounds mainly include decanal, (z)-2-decenal, (e)-2-undecenal, and octanal. Among them, decanal has a flowery and fruity odor, (z)-2-decenal has a fruity or fat odor, (e)-2-undecenal has a cabbage odor, and octanal has a citrus or green odor. These furans and alkenals may also have affected the aromas of tea samples of different suitability.

In this study, it was shown that hs-spme-gc-ms could significantly distinguish the content differences of aroma compounds between the products of tea cultivars differing in suitability for making oolong tea, with 47 aroma compounds being significantly different between the two groups. If we adopt more advanced analytical means in future (eg gc-ims), more different substances may be obtained according to this work [25]. Some aroma compounds in oolong tea (such as indole, nerolidol, farnesene, etc.) are synthesized de novo in the withering process due to mechanical damage or low temperature environment [3,23]; other compounds (such as furans, fatty aldehydes, etc.) belong to thermochemical reaction products, which are mainly synthesized during thermal processing, such as in fixing and drying after withering [26]. Therefore, the aroma difference in oolong tea cultivars of different suitability can be attributed to the differences in the synthesis of related substances in the two aforementioned processes.

### 3.4. Association Analysis and Discussion

The above results indicate that the differences in the types and contents of aroma substances may be the main reasons for the differences in the suitability of cultivars, but do these aroma substances affect human sensory evaluation? This suggests the necessity of further analysis of their association, so Spearman’s correlation analysis was used to analyze the relationships among aroma substance data, sensory data, and electronic nose data. Figure 4 and Appendix A show the correlations among 47 significantly different aroma substance data, 16 electronic nose sensor responses values, and consumer acceptance and expert review scores.

Among volatile terpenes, Cosmene, Linalool oxide pyranoside, α-Cubebene, β-Caryophyllen, α-Muurolene, δ-Cadinene, α-Cadinol, and 16 electronic nose sensors (LY type sensors: LY2/AA, LY2g/CTL, LY2g/CT, LY2/G, and LY2/GH; T type: TA/2, T70/2, T40/2, and T30/1; *p* type: P30/2, P10/2, PA/2, P30/1, P40/2, P40/1, and P10/1) all had weak correlation (0.3 < /correlation coefficient/ < 0.5), no correlation (/correlation coefficient/ < 0.3), or no strong correlation (/correlation coefficient/ < 0.7) with human sensory data (expert review data and consumer acceptance data). This indicates that these seven terpenes are likely to have little effect on the human sensory perception of samples from different cultivars. α-Terpineol, Linalyl formate, Geraniol, α-Farnesene, (E)-Nerolidol, (E)-Furan linalool oxide, Linalool oxide pyranoside, and T-Muurolol had a weak correlation (/correlation coefficient/ < 0.5) with LY type sensors, but a moderately strong correlation (0.5 < /correlation coefficient/ < 0.7) or strong correlation (0.7 < /correlation coefficient/ < 1) with almost all T type and *p* type sensors. It is worth noting that α-Terpineol, Geraniol, (E)-Nerolidol, and α-Farnesene showed weak correlation (0.3 < /correlation coefficient/ < 0.5) or no correlation (0 < /correlation coefficient/ < 0.3) with human sensory data. This indicated that these four substances did not contribute to the sensory differences in samples of different suitability. However, β-Ocimene, (Z)-Furan linalool oxide, linalool, (3E)-4,8-Dimethyl-1,3,7-nonatriene, (E)-Pyranoid linalool oxide, γ-Elemene, Humulene, (E)-β-Famesene, and (Z,E)-α-Farnesene not only had a strong correlation with the electronic nose array, but also significantly affected the sensory score level, suggesting that they may have contributed to the sensory differences in samples from cultivars of different suitability. As metabolites present widely in a variety of plants, terpenes are not only the constituents of the floral fragrance released by plants, but also play a role in plant defense under biotic and abiotic stress [27,28]. In tea leaves, α-Farnesene and Ocimene have been found to be synthesized de novo in tea leaves treated with jasmonic acid and mechanical damage (JAMD) and pass between tea plants to initiate volatile defense mechanisms of undamaged tea plants [29]. In another report, CsOCS was mentioned as a key synthase to promote the de novo synthesis of β-ocimene in oolong tea processing, which is one of the important defense signals of the tea plant against adversity [30]. These reports suggested that although some of the aforementioned terpene substances (such as α-Farnesene, etc.) did not contribute to human sensory perception, they may be involved in the tea tree defense signaling network in response to adverse processes (mechanical damage and low temperature processes) and indirectly affect the formation of other aroma components, and so are some of the substances that contribute to sensory formation (such as β-Ocimene, etc.). This may be the main reason why these terpenes caused the sensory aroma differences in samples of different suitability.

Among amino acid derived volatiles, Methyl salicylate, Geranyl isovalerate, and indole not only showed strong correlations with sensor arrays, but also had moderate or strong correlations with sensory evaluation results. Indole, one of the characteristic components of oolong tea, was confirmed here as a differential aroma compound contributing to oolong tea suitability. Unexpectedly, methyl salicylate was confirmed as one of the important characteristic volatiles contributing to the floral aroma of Keemun black tea in a previous study [31], and its content can be used to characterize the fermentation degree of tea [21,32]. Here, methyl salicylate was also found to be a differential substance affecting the suitability of oolong tea, which suggests that it may be involved in the formation of the floral aroma of cultivars suitable for making oolong tea.

Among fatty acid derived volatiles, (Z)-3-Hexenyl (Z)-3-hexenoate, Butanoic acid hexyl ester, and (Z)-Jasmone had strong correlations with electronic nose data and sensory evaluation data, indicating that these three compounds contributed to the suitability of oolong tea cultivars. Butanoic acid hexyl ester is one of the characteristic active compounds of osmanthus aroma, with a violet, woody, and fruity odor [33]. (Z)-Jasmone is one of the characteristic aromas of oolong tea [21], and also one of the main components of the plant’s direct or indirect stress-resistance mechanism [34]. Despite rare reports on the direct or indirect involvement of (Z)-Jasmone in the anti-stress effect of tea, cotton was shown to release terpenes (β-ocimene, DMTT, TMTT) by exogenously adding (Z)-Jasmone to achieve stress resistance mechanism [35], which suggests that (Z)-Jasmone may be involved in the regulation of aroma compounds during oolong tea processing (mechanical damage and low temperature). Different aroma compounds produced by cultivars of different suitability in response to adversity may contribute to the sensory differences of oolong tea. (Z)-3-Hexenyl (Z)-3-hexenoate, a hexanoic acid moiety-containing compound belonging to green leaf volatiles (GLVs), was reported to be related to the role of jasmonic acid in plant stress [36,37].

Among carotenoid derived volatiles, α-Ionone had a low correlation with sensor response values and sensory scores, suggesting that it might not participate in the formation of differential senses. On the contrary, β-Ionone, (Z)-Geranylacetone, and 6-methyl-5-Hepten-2-one were strongly correlated with electronic nose sensor response values and sensory scores, suggesting that these substances may contribute to the formation of different aroma. β-Ionone, a floral substance with a low threshold value, is one of the characteristic components of floral aroma in black tea [38]. Carotenoid cleavage enzymes (CCD1) were reported to cleave the 5’6’ double bond of lycopene to obtain 6-methyl-5-Hepten-2-on [14]. In a tomato study, CCD1 was shown to cleave the 9’10’ double bond of cyclic or acyclic carotenoids to produce β-Ionone and (Z)-Geranylacetone [39]. These results indicate that the differences of β-Ionone and (Z)-Geranylacetone in samples of different cultivars may be affected by the expression level of CCD1 enzyme, thus affecting the cultivar suitability.

Among fatty aldehydes and furan compounds, Octanal, Decanal, (E, E)-2,4-Nonadienal, (Z)-2-Decenal, (E)-2-Undecenal, and 2-Pentylfuran had a strong correlation with the sensor response value and sensory scores, suggesting these compounds may significantly affect a cultivar’s suitability for oolong tea.

Based on the above analysis, it can be concluded that the aroma differences in oolong tea samples from cultivars of different suitability were jointly affected by eight terpene metabolites (β-Ocimene, (Z)-Furan linalool oxide, linalool, (3E)-4,8-Dimethyl-1,3,7-nonatriene, (E)-Pyranoid linalool oxide, γ -Elemene, Humulene, and (Z,E)-α-Farnesene), three carotenoid metabolites (β-Ionone, (Z)-Geranylacetone and 6-methyl-5-Hepten-2-one), three lipid metabolites (cis -3-Hexenyl (Z)-3-hexenoate, Butanoic acid hexyl ester, and (Z)-Jasmone), four amino acid metabolites (Methyl salicylate, Geranyl isovalerate, indole, Phenylethyl alcohol), and six thermal reaction products (2-Pentylfuran, Octanal, Decanal, (E, E)-2,4-Nonadienal, (Z)-2-Decenal, and (E)-2-Undecenal).

## 4. Conclusions

This study confirmed significant differences in aroma between traditional suitable and unsuitable tea cultivars for oolong tea processing based on sensory data, electronic nose data, HS-SPME-GC-MS data and their correlations. Expert review and consumer acceptance evaluation scores were consistent with the suitability classifications. The electronic nose data adequately distinguished the suitability classification, albeit with some blind spots. HS-SPME-GC-MS data and correlation analysis suggested that some volatiles (such as (Z)-Jasmone and β-Ocimene) may affect the sensory differences in tea cultivars of different suitability, and are related to the characteristics of the cultivar, thus being termed as “Cultivar aroma”. Several other aroma compounds with a high content (such as (E)-Nerolidol and α-Farnesene) in oolong tea are generated in the processing mode; however, they are weakly related to cultivar suitability and are termed as “Technical aroma”. Both cultivar aroma and technical aroma may contribute jointly to the aromatic characteristics of oolong tea. This work preliminarily interprets the relationship between the aroma and the suitability of oolong tea, and enlightens the research on the suitability of tea cultivars. The findings provide a theoretical basis for the issue of cultivar selection for global tea production.

## Figures and Tables

**Figure 1 foods-11-02880-f001:**
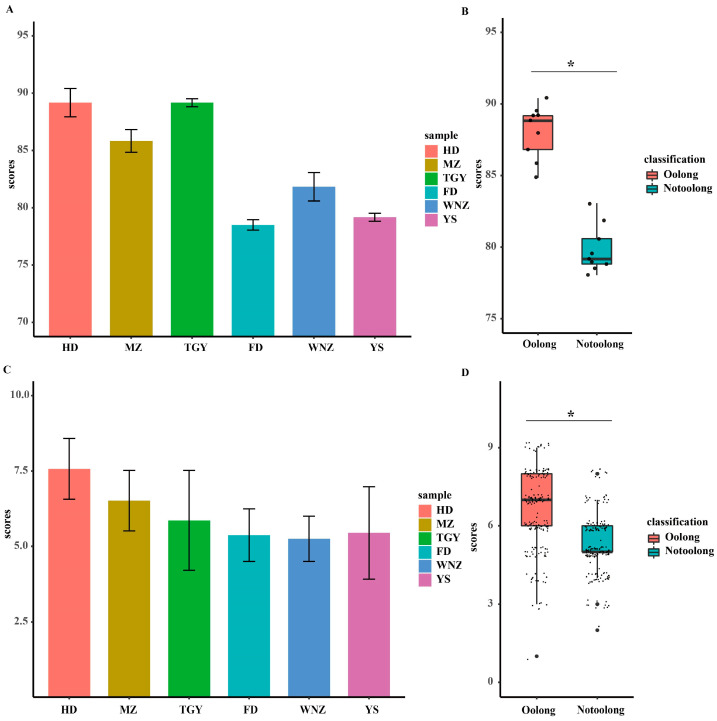
Sensory evaluation score plot for samples of six different tea cultivars. (**A**) Histogram of scores from expert sensory evaluation for the six different tea samples (HD, MZ, TGY, FD, WNZ, and YS). (**B**) Boxplot of expert sensory evaluation scores of oolong tea samples made from suitable cultivars (HD, MZ, TGY) and unsuitable cultivars (FD, WNZ, YS). (**C**) Histogram of consumer acceptance evaluation scores of six tea cultivars. (**D**) Boxplot of consumer acceptance evaluation scores of oolong tea made from suitable cultivars (HD, MZ, TGY) and unsuitable cultivars (FD, WNZ, YS). Oolong: oolong tea samples made from suitable cultivars (HD, MZ, TGY); Notoolong: oolong tea samples made from unsuitable cultivars (HD, MZ, TGY). “*” indicates significant difference at *p <* 0.01 in Student t-test. The black dots: the scores from every expert and consumer.

**Figure 2 foods-11-02880-f002:**
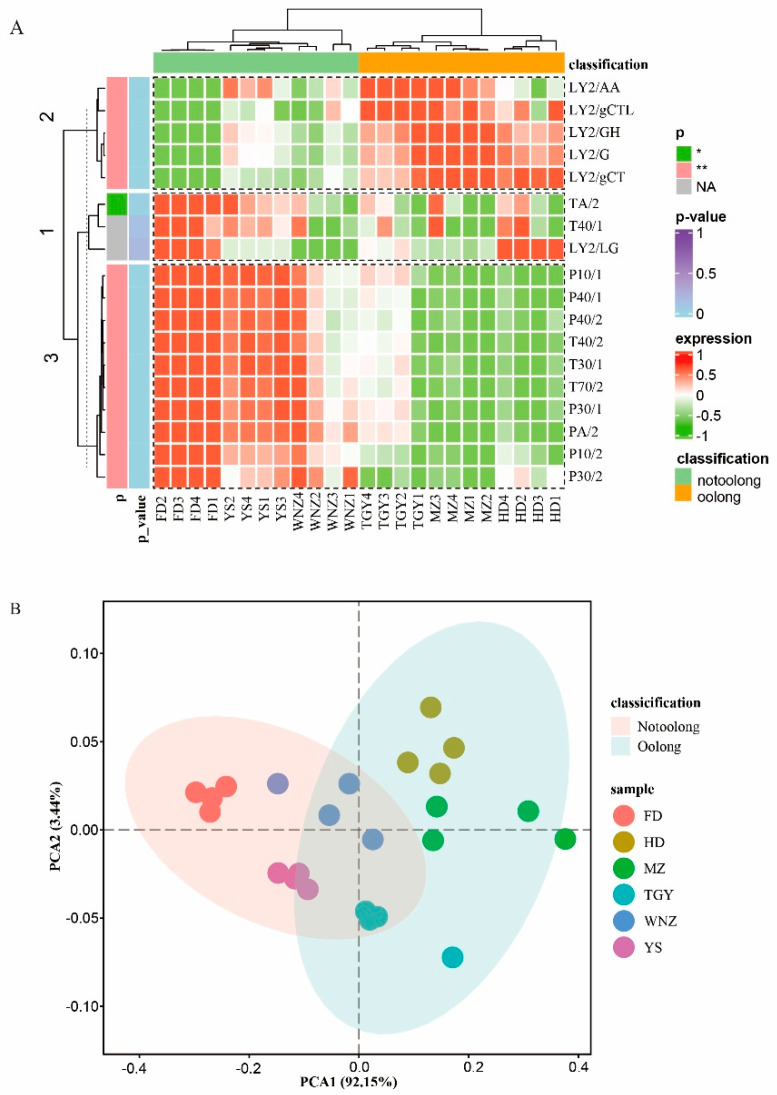
Stable response values of electronic nose sensors for samples of six different tea cultivars. (**A**) Heat map of electronic nose response values for samples of six different tea cultivars. (*p*: significant results based on Kruskal–Wallis test; “*” indicates *p <* 0.05; “**” indicates *p <* 0.01, and “NA” indicates *p* > 0.05. expression: the size of the response values of different samples to the same sensor. (**B**) The PCA analysis score plot of electronic nose response values for six different tea cultivars. (grouping method: ggplot2::stat_ellipse, confidence interval: 0.95). Oolong: oolong tea samples made from suitable cultivars (HD, MZ, TGY); Notoolong: oolong tea samples made from unsuitable cultivars (HD, MZ, TGY).

**Figure 3 foods-11-02880-f003:**
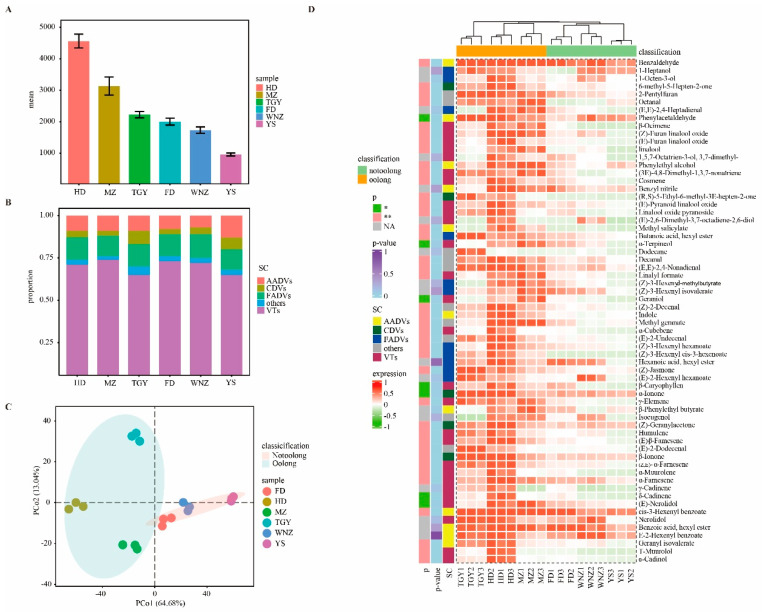
Aroma characteristics of samples from six different tea cultivars detected by HS-SPME-GC-MS. (**A**) HS-SPME-GC-MS bar graph of total aroma in the samples of six different tea cultivars. (**B**) Aroma classification stacking map of samples from six different tea cultivars measured by HS-SPME-GC-MS. (**C**) PCoA analysis score plot of HS-SPME-GC-MS results for samples from six different tea cultivars. (Grouping method: ggplot2::stat_ellipse, confidence interval: 0.95). (**D**) Heat map of HS-SPME-GC-MS results of six different tea cultivars. p_value: the left annotation heatmap color represents the size of the p-value data based on the Kruskal-–Wallis test. *p*: significant results based on Kruskal–Wallis test; “*” means *p <* 0.05; “**” means *p <* 0.01; “NA” means *p* > 0.05. SC: classification of substances according to different metabolic pathways; classification: clustering results between sample groups using the Spearman method; expression: relative content of the same substance in different samples. Oolong: oolong tea samples made from suitable cultivars (HD, MZ, TGY); Notoolong: oolong tea samples made from unsuitable cultivars (HD, MZ, TGY). Detailed information is also presented in Appendix A. Since this study focused on suitability differences between the six oolong tea cultivars, PCoA analysis was performed to further examine the aroma contents of six oolong tea cultivars measured by HS-SPME-GC-MS. In the PCoA analysis score chart (Figure 3C), the principal coordinate components PCoA1 and PCoA2 explained more than 75% of the inter-group variation in this model, and the inter-sample distribution of aroma content was significant in the six tea samples (R^2^ = 0.97109, *p <* 0.01) (Appendix A). In Appendix A, Adonis analysis results of sample grouping showed the significant aroma differences between the groups of suitable and unsuitable tea cultivars (MZ, HD, TGY versus FD, WNZ, YS) (*p <* 0.01, R^2^ = 0.51893), further suggesting data reliability. The grouping results here agreed well with the above sensory review data and electronic nose data, suggesting a correlation between the three datasets.

**Figure 4 foods-11-02880-f004:**
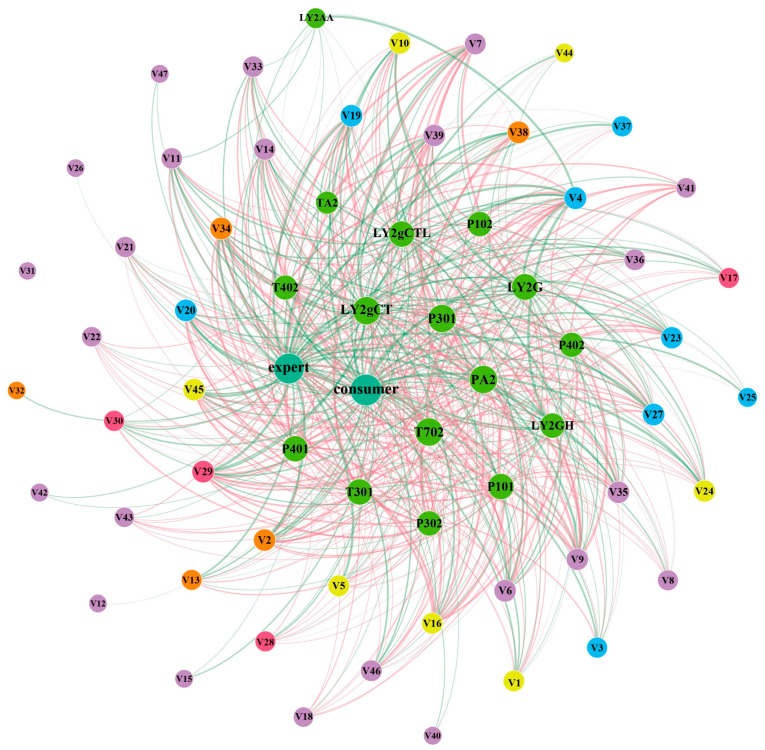
Network diagram of the relationship between electronic nose data, HS-SPME-GC-MS data, and sensory data in six tea cultivars. Node information: V1: Benzaldehyde; V2: 6-Methyl-5-Hepten-2-one; V3: 2-Pentylfuran; V4: Octanal; V5: Phenylacetaldehyde; V6: β-Ocimene; V7: (Z)-Furan linalool oxide; V8: (E)-Furan linalool oxide; V9: Linalool; V10: Phenylethyl alcohol; V11: (3E)-4,8-Dimethyl-1,3,7-nonatriene; V12: Cosmene; V13: (R,S)-5-Ethyl-6-methyl-3E-hepten-2-one; V14: (E)-Pyranoid linalool oxide; V15: Linalool oxide pyranoside; V16: Methyl salicylate; V17: Butanoic acid hexyl ester; V18: α-Terpineol; V19: Decanal; V20: (E,E)-2,4-Nonadienal; V21: Linalyl formate; V22: Geraniol; V23: (Z)-2-Decenal; V24: Indole; V25: Methyl geranate; V26: α-Cubebene; V27: (E)-2-Undecenal; V28: (Z)-3-Hexenyl hexanoate; V29: (Z)-3-Hexenyl (Z)-3-hexenoate; V30: (Z)-Jasmone; V31: β-Caryophyllen; V32: α-Ionone; V33: γ-Elemene; V34: (Z)-Geranylacetone; V35: Humulene; V36: (E)-β-Famesene; V37: (E)-2-Dodecenal; V38: β-Ionone; V39: (Z,E)-α-Farnesene; V40: α-Muurolene; V41: α-Farnesene; V42: δ-Cadinene; V43: (E)-Nerolidol; V44: (Z)-3-Hexenyl benzoate; V45: Geranyl isovalerate; V46: T-Muurolol; V47: α-Cadinol. Connection information: the red line segment represents negative correlation; the green line segment, positive correlation; line segment thickness, correlation magnitude.

## Data Availability

Data is contained within the article.

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
