# Peer review of "Study on the Suitability of Tea Cultivars for Processing Oolong Tea from the Perspective of Aroma Based on Olfactory Sensory, Electronic Nose, and GC-MS Data Correlation Analysis"

_foods, 2022, doi:10.3390/foods11182880_

Round 1

Reviewer 1 Report

The authors performed a study to evalauate the differences between oolong teas from different cultivars by using olfactory sensory, electronic nose and GC-MS data correlation analysis. 

First of all i would like to thank for this valuable study, i find it quite important and novel.

Please see the pdf document with notes on it. Please perform questions on the basis of these notes. 

I found the studies below with similar concepts please highlight the superiority and differences of your research in the discussion part when compared to previous ones.

1)Oolong tea made from tea plants from different locations in Yunnan and Fujian, China showed similar aroma but different taste characteristics

2) Characterization of the aroma profiles of oolong tea made from three tea cultivars by both GC–MS and GC-IMS

Conclusion: Please highlight the importance and benefits of your findings globally.

Please consider professional English edit.

Reviewer 2 Report

Dear Authors,

In general, the manuscript is good to read, the structure of the work is clear and has a sufficient literature review. I present my comments below:

1.      Please check the entire manuscript for no spaces before citation numbers.

2.      1. Introduction. In order to extend the literature review, the authors may mention that electronic noses are also used to distinguish between coffee species, oil testing etc ... Some examples of new works: "Impact of Coffee Bean Roasting on the Content of Pyridines Determined by Analysis of Volatile Organic Compounds, Impact of Coffee Bean Roasting on the Content of Pyridines Determined by Analysis of Volatile Organic Compounds" or "Identification of changes in the volatile compounds of robusta coffee beans during drying based on HS-SPME/GC-MS and E-nose analyses with the aid of chemometrics" or "Identification of the olfactory profile of rapeseed oil as a function of heating time and ratio of volume and surface area of contact with oxygen using an electronic nose".

3.      2.4. Electronic nose detection. Has the tea data library been used?

4.      2.5. Extraction and analysis of volatile compounds in oolong tea. On what basis were the column and SPME fiber selected?

5.      Figure 3D is hardly legible.

 In addition, the work appears complete and almost ready for publication.
